# Optimization of Dynamic Task Location within a Manipulator's Workspace for the Utilization of the Minimum Required Joint Torques

Adam Wolniakowski [1], Charalampos Valsamos [2], Kanstantsin Miatliuk [3,*], Vassilis Moulianitis [4] and Nikos Aspragathos [2]

[1] Faculty of Electrical Engineering, Bialystok University of Technology, 15-351 Bialystok, Poland; a.wolniakowski@pb.edu.pl
[2] Department of Mechanical Engineering and Aeronautics, Polytechnic School, University of Partas, 26500 Patras, Greece; balsamos@upatras.gr (C.V.); asprag@mech.upatras.gr (N.A.)
[3] Department of Mechatronics and Robotics, Faculty of Mechanical Engineering, Bialystok University of Technology, 15-351 Bialystok, Poland
[4] Department of Product and System Design Engineering, School of Engineering, University of the Aegean, 84100 Syros, Greece; moulianitis@syros.aegean.gr
[*] Correspondence: k.miatliuk@pb.edu.pl; Tel.: +48-571-443-057

**Abstract:** The determination of the optimal position of a robotic task within a manipulator's workspace is crucial for the manipulator to achieve high performance regarding selected aspects of its operation. In this paper, a method for determining the optimal task placement for a serial manipulator is presented, so that the required joint torques are minimized. The task considered comprises the exercise of a given force in a given direction along a 3D path followed by the end effector. Given that many such tasks are usually conducted by human workers and as such the utilized trajectories are quite complex to model, a Human Robot Interaction (HRI) approach was chosen to define the task, where the robot is taught the task trajectory by a human operator. Furthermore, the presented method considers the singular free paths of the manipulator's end-effector motion in the configuration space. Simulation results are utilized to set up a physical execution of the task in the optimal derived position within a UR-3 manipulator's workspace. For reference the task is also placed at an arbitrary "bad" location in order to validate the simulation results. Experimental results verify that the positioning of the task at the optimal location derived by the presented method allows for the task execution with minimum joint torques as opposed to the arbitrary position.

**Keywords:** optimization; human–robot interaction; dynamic task; simulation; universal robot

## 1. Introduction

During the planning of a robotic task, the designer must address a number of key issues regarding the task execution by the manipulator, so as the resulting workcell will achieve the best possible performance. One of the cornerstones in robotic task design is the determination of the workcell configuration and most importantly, the positioning of the task within the manipulator's workspace so that during task execution the robot would be able to exert high performance characteristics as per the task requirements. Such characteristics may vary, depending on the type of task (kinematic or dynamic) as well as on the requirements regarding performance as set by the designer (i.e., minimization of cycle time, low energy consumption etc.). Other aspects regarding the physical configuration of the workcell may also present further constraints, such as maximum joint velocity and acceleration, maximum accuracy (deviation tolerance) and obstacle avoidance, or process constraints such as time window or end-effector velocity and acceleration limits. In an era where there is a general strive towards green manufacturing, robotic tasks, which in many cases are quite energy intensive, should also be considered for optimization under such a

context. Due to the direct proportional relationship between the manipulator's joint torques and the consumed energy during the exertion of these torques by the joints, minimizing the required torques required for the task would inevitably reduce the manipulator's power consumption.

The problem of optimizing a task in order to minimize the required joint torques and subsequently the consumed energy by a manipulator has already been addressed to a degree in the relevant literature. In general, this is achieved either by placing the task at an optimal location within the manipulator's workspace or by utilizing an accordingly designed control scheme. A motion control scheme for multi-joint robots using elastic elements with adjustable stiffness installed to each joint was presented in [1]. The motion control scheme adjusted the stiffness of these elements in an adaptive manner, minimizing the actuator torques without the requirement of the a priori knowledge of the exact values of physical parameters or elaborate numerical calculations. The presented experimental results showed a significant reduction in the required joint torques by the manipulator.

Similarly, a control method for joint torque minimization of redundant manipulators handling large external forces was presented in [2]. The presented method utilized null space control in order to design the task so as to minimize the torques required to oppose the external force acting on the end effector and reduce the dynamic torque. As such the proposed control method took into account the externally applied forces to the robot's end effector as well as the internal dynamics of the manipulator. The proposed control method was verified through two different case studies, a simulation of high-pressure blasting and that of a manipulator lifting and moving a heavy object. The presented results showed an overall reduction in joint torques compared to conventional methods. Furthermore, the joint torque was minimized such that there was a potential for manipulator to execute certain tasks beyond its nominal payload capacity.

In other proposed methods, the task is optimized both kinematically and dynamically. For instance, in [3], a method for the different-level simultaneous minimization of joint-velocity and joint-torque for redundant robot manipulators was presented. The method combined the minimum two norm joint-velocity and joint-torque solutions via two weighting actors. Physical constraints such as joint-angle limits, joint-velocity limits and joint acceleration limits ware also taken into consideration. The presented simulation results were based on the PUMA560 robot manipulator performing different types of end-effector path-tracking tasks demonstrating the validity and advantage of the proposed different level simultaneous minimization scheme. Furthermore, experimental verification conducted on a practical six-link planar robot manipulator substantiated the effectiveness and the physical feasibility of the proposed scheme.

In [4], an approach for the development of an optimization module that minimizes the energy consumption of the robot's movement was presented. The authors solved the inverse kinematics and the inverse dynamics of the robot at each configuration during task execution, while the energy consumption was calculated for each configuration. The module selected the optimal configuration of the joint variables and joint torques that minimized the power consumption during task execution and provided the data to the robot controller. Three case studies were used to evaluate the performance of the module and experimental results demonstrated the developed module as a successful tool for energy efficient robot path planning. Further analyses for the results had been done by comparing them with the ones from commercial simulation software. The case studies showed that the optimization of the location for the target path could reduce the energy consumption effectively.

In a similar fashion, a method for reducing the energy consumption of pick-and-place industrial robots was presented in [5]. The authors derived electromechanical models of both serial and parallel manipulators and utilized them to calculate the energy-optimal trajectories, by means of constant time scaling, starting from pre-scheduled trajectories compatible with the actuation limits. In this manner, the robot work cycle was energetically optimized also when the TCP (Tool Center Point) position profiles had been already

defined on the basis of technological constraints and/or design choices aimed at ensuring manufacturing process efficacy/robustness. The effectiveness of the proposed procedure was finally evaluated on two simulation case studies.

In [6], a method for the optimal task placement of a serial robot manipulator for manipulability and mechanical power optimization, was presented. The end-effector was constrained to follow a predefined path during the optimal task positioning. The proposed strategy defined a relation between mechanical power and manipulability as a key element of the manipulator analysis, establishing a performance index for a rigid body transformation. This transformation was used to compute the optimal task positioning through the optimization of a multi-criteria objective function. Numerical simulations regarding a serial robot manipulator demonstrated the viability of the proposed method.

However, the undertaking of tasks by manipulators in various sectors, in the place of a human operator or worker, provides additional challenges for task design and placement, due to the complexity of the required motions and trajectories to be followed by the manipulator's end effector. Such trajectories are in many cases difficult to model computationally. As such, it is advantageous if the robot was to be "taught" the required trajectory to be followed by the human operator. For example, Kand and Ikeuchi presented a method for the programming of a manipulator via the observance of a human operator performing a given task [7]. Similarly, Wei et al. in [8] presented a similar method of teaching a manipulator a given task to be performed via the observance of a human operator conducting the task. A similar method was also proposed in [9] where the authors used the sensors of a mobile phone and a Microsoft Kinect sensor to provide the manipulator with the required position and orientation of the end effector at given task points.

In this paper, a method for the optimal placement of a dynamic robotic tasks within a manipulator's workspace is presented. Although the approach presented shares similar concepts to the works cited, there are two significant impact points that are significantly different. The first one, is the determination of the dynamic task location taking into account the minimum singularity free paths that the manipulator's end effector can transverse to execute the task. As such, the derived location ensures that the task will be executed without the possibility of the manipulator reaching a singular posture. In [7] such a consideration was presented for the design of a kinematic robotic task, in the used multi criteria formulation, however in a global fashion—within the whole workspace. In this paper, singularity avoidance is ensured within a dynamic task-based fashion, which is the main contribution of this work. The second one concerns the evaluation and verification of the proposed methodology for the optimal task placement experimentally. Due to most offline processes presented in the literature being mostly evaluated through simulation, the experimental evaluation allows better evaluation of the proposed method. As such it is possible to determine whether the proposed method ensures that the derived task location allows for the manipulator's high level dynamic performance during task execution.

The presented method allows for the derivation of the optimal task location in a manipulator's workspace, where joint torques required to execute the task with the end effector exerting a given force along the considered task path are minimized. A task-based performance index is proposed based on the manipulator mechanical advantage (MMA) [10]. The proposed method aims at the placement of the task in the manipulator's workspace at a location where the maximization of the minimum value of the proposed task index is achieved. The result of this design stage, i.e., the task location, is subsequently used as input to an experimental implementation set up, where the task is executed. For the purpose of application of the method the application of a force by the end effector along a trajectory is considered. An experimental set-up is used to undertake the task execution using a UR-3 serial manipulator [11]. Additionally, due to the complex nature of the task path considered in the case of the experimental application of the proposed method, a human–robot interaction aspect and process was considered and utilized, with the manipulator being taught the trajectory by a human operator. The derived optimal as well as a "bad" location for each of the tasks are considered and the joint currents, and task

execution time, are measured during task execution. The results of the two cases for each task are then compared to determine the validity of the presented method.

## 2. The Proposed Optimization Method

In general, during a dynamic robotic task, the manipulator is to reach a number of target points that can be either in specific locations or along a given path, located within its workspace and with a given orientation of its tool frame. In the case of dynamic tasks, these points are located in areas where the tool at the manipulator's end effector performs a specific function by exerting a given force (such tasks may be pick and place tasks, spot welding, etc.). Alternatively, these points may set points along a given path that the end effector tool must follow and exert a given force along the whole path, as defined by the task's specifications (as for example in welding applications, applying glue on surfaces, etc.).

To obtain high dynamic performance during its execution, the task must be placed at the optimal location in the manipulator's workspace [1–6]. For the presented method, dynamic tasks, are considered to be defined by a number of target points to be visited with a given orientation by the robot's end effector, where a specific force/torque must be applied.

The process of the method realization consists of two interconnected parts, the layout design and the experimental implementation, as illustrated in Figure 1.

In the layout design part, the optimal location of the task is determined. The process inputs are the set of the target points describing the task and the required force to be applied at them by the end effector. A Local Coordinate System {L} is placed at one of the target points, depicting the location and the orientation of the tool frame when it reaches the specific point, while the relative coordinate systems placed at the remaining target points are referenced to that. As such the target points are defined by the position and orientation of their local coordinate systems that the manipulator's end effector must reach, relative to {L}, and the sequence in which the end effector tool must approach them.

The first step of the layout design process involves the generation of intermediate coordinate frames along the paths between every couple of consecutive points. The number of intermediate frames generated upon each path is decided by the designer, given the task specifications. Upon the completion of the process a set of task coordinate frames is created, including those placed at the task's target points.

Secondly, the derivation of the best paths between consecutive task frames takes place. A task-based dynamic index is used to provide a proportional indication of a selected aspect of robot dynamic performance (such as the MMA). For a motion between two consecutive target frames, this procedure involves the following sub steps:

- the inverse kinematics problem is solved at each task frame along the path to provide the possible configurations for the manipulator to reach each frame (for six Degrees of Freedom (DoF) manipulators there are a maximum of eight configurations per frame);
- all possible singular free paths for the motion between two consecutive task frames are generated in the configuration space, using the combinations of the inverse kinematics solutions derived at the previous step;
- the dynamic index value is calculated for every singular free path;

Based on the index value along each path and the optimization requirements (maximization, minimization) the best path is determined (objective function). The process is employed in a genetic algorithm (GA) which is used to conduct a search for the optimal location of the dynamic task.

The GA utilizes the best path derivation process for each placement of the dynamic task within the manipulator's workspace. At the end of the search process the optimal location of the dynamic task is determined.

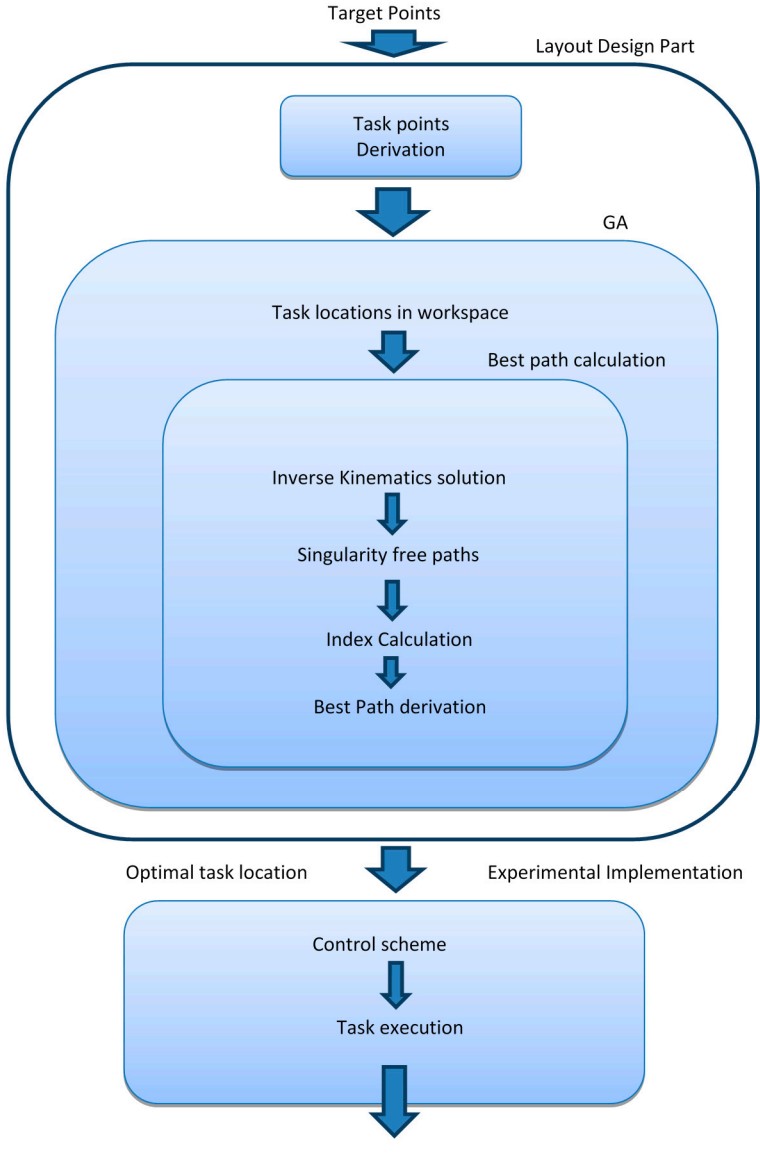

**Figure 1.** Process of the proposed method realization.

In the experimental implementation part, the input is the determined optimal position and orientation of {L} with respect to {B} (Base Coordinate System of the Robot), the referencing of the remaining task frames to {L} and the sequence with which the end effector must approach them. These are used to configure the physical execution of the task by a robotic work cell, in order to measure the robot's performance. In this work the workcell was composed of a UR-3 six-DoF manipulator and the necessary fixtures for the task's execution.

## 2.1. Formulation of the Optimal Task Placement Problem

The manipulator is to reach k frames with given orientation and at given distances between each other formulating the desired robotic task. In most industrial robotic tasks, whether kinematic or dynamic, the positional and orientational relationship between the task frames is a priori known. For example, the application of adhesive for the placement of the windscreen on an automobile chassis is depicted as a known path along the edge of the opening where the part is to be placed. As such the relative position and orientation of each frame with respect to the others is fixed and known. Therefore, in the case of task planning this provides an advantage since a single frame can be arbitrarily chosen where a frame

depicting the task may be placed. Since the remaining task frames are easily referenced to the task frame, the problem can be therefore simplified to the determination of the optimal location of the task frame.

A graphical representation of the considered problem is presented in Figure 2. The robot is presented at an arbitrary configuration, while the set of task frames representing the desired end effector tool positions and orientations that compose the task are represented as $\{\mathbf{p}_i\}$, $i = 2, \ldots, k$. In Figure 2 the base coordinate system of the robot (BCS) is denoted as **{B}**, the end effector's tool frame (TCS) is denoted as **{T}** and the local coordinate system of the task (LCS) is denoted as **{L}**, are presented, where $\{\mathbf{L}\} \equiv \{\mathbf{P_1}\}$. The task frames positions and orientations relative to **{L}** are known and given by:

$$^{L}\mathbf{M}_i = \begin{bmatrix} & ^{L}\mathbf{R}_i & & ^{L}\mathbf{p}_i \\ 0 & 0 & 0 & 1 \end{bmatrix}, \; i = 2 \ldots k, \tag{1}$$

where $^{L}\mathbf{R}_i$ is the $3 \times 3$ matrix representing the orientation of the $i^{\text{th}}$ frame coordinate system relative to **{L}** and $^{L}\mathbf{p}_i$ is the $3 \times 1$ vector representing the location of the $i^{\text{th}}$ frame relative to **{L}**. The location of the task is therefore accordingly depicted by the position and orientation of **{L}** relative to **{B}**:

$$^{B}\mathbf{M}_L\left(^{B}\mathbf{p}_{L}, \underline{\theta}\right) = \begin{bmatrix} ^{B}\mathbf{R}_L(\underline{\theta}) & ^{B}\mathbf{p}_L \\ 0 \;\; 0 \;\; 0 & 1 \end{bmatrix}, \tag{2}$$

where $\underline{\theta}$ are the z-y-z Euler angles depicting the orientation of the frame relative to **{B}**. The problem of the optimal placement of the task points, considering this definition is transformed to the optimal placement problem of **{L}** relative to **{B}**.

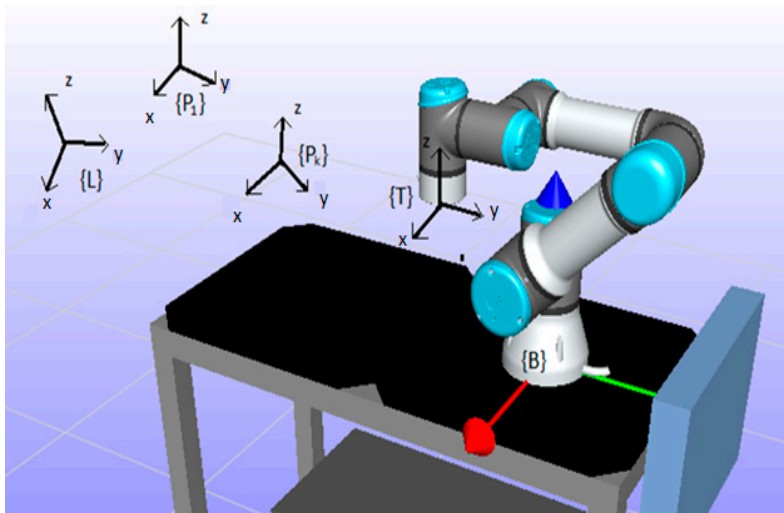

**Figure 2.** UR3 manipulator and the base **{B}**, end effector's **{T}**, the local **{L}** coordinate systems, and the task frames $\{\mathbf{p}_i\}$.

The task frames are used to determine $k$ motion segments representing the task path where the orientation of the end effector as well as the force exerted by it are considered to be constant. A main requirement for desired tasks is usually the placing of the task in such a location in the manipulator's workspace where its end effector will be capable of exerting the required force at the task frames utilizing the minimum possible torques of its joints. As such, two different aspects of the manipulator's dynamic performance may be achieved: (a) the maximization of the force the end effector is capable of applying on the task points, given the joint torques limits and (b) the minimization of the power consumption by the manipulator to complete the task [5]. Given the above requirement and optimization target, the local performance index chosen in order to determine the optimal location of the task in the manipulator's workspace is the Manipulator Mechanical Advantage (MMA) [11].

The MMA is a measure of the transmission of the joint torques to the forces applied by the end effector at a given location and direction. As such, a high value of the index signifies that the robot can exert high forces at its tool frame, utilizing low joint torques.

Therefore, the calculation of the MMA requires the knowledge of the direction of the force applied by the end effector. The direction of the exerted force along the path is usually specified by the task parameters. Therefore, the value of the MMA may be calculated for each motion segment, using the direction of the applied force from the starting task frame and the configurations of the robot depicting the solution to its inverse kinematics problem at the starting frame. As such, the MMA value for the motion along two consecutive frames is given as:

$$r_{mi,j}\left({}^{B}\mathbf{p}_{L}, \underline{\theta}\right) = \cfrac{1}{\sqrt{\mathbf{u}_{mi}^{T}\left[\left(J_{mi,j}\left({}^{B}\mathbf{p}_{L}, \underline{\theta}\right)\right)\left(J_{mi,j}\left({}^{B}\mathbf{p}_{L}, \underline{\theta}\right)\right)^{T}\right]\mathbf{u}_{mi}^{-1}}}, \; i = 1,\dots,k, \; j = 1,\dots 8, \tag{3}$$

where $j$ is the index for the corresponding inverse kinematics solution for the $i^{\text{th}}$ point (max eight solutions for a typical six-DoF articulated manipulator with a spherical wrist). Vector $\mathbf{u}_{mi}$ is the unit vector depicting the direction of force exerted by the end effector for the given segment. However, the different solutions of the inverse kinematics to the $i^{\text{th}}$ and $(I+1)^{\text{th}}$ frames define a number of paths in the joint space that the manipulator may use to perform the motion between them. As such, the shortest singular free path is sought for each motion segment.

### 2.1.1. Singular Free Paths in Configuration Space

According to [12] no-generic geometries divide the space of 3R geometries into disjoint sets which have homogeneous topological properties. Four disjoint sets can be produced from a 3R geometry containing the four solutions of the inverse kinematic problem (IKP). Therefore, the IKP solutions in non-cuspidal manipulators are separated by the singularity locus, and during the change of postures the manipulator will have to pass through a singular point. The determination of the disjoint sets and their correspondence to the inverse kinematics solutions depends on the determinant of the Jacobian matrix $|J|$. It has been shown that the determinant of a manipulator does not depend on the first joint variable [13]. In this paper, the UR3 manipulator is a six-DoF manipulator with eight solutions to the inverse kinematics problem. The determinant of the UR-3 manipulator is:

$$|J| = -\frac{L_2 L_3 \sin q_5}{2}\left(d_5 \cos(q_2 + 2q_3 + q_4) - L_2 \sin(q_2 + q_3) - d_5 \cos(q_2 + q_4) + L_3 \sin q_2 + L_2 \sin(q_2 - q_3)\right) \\ - L_3 \sin(q_2 + 2q_3) \tag{4}$$

where, $q_2, q_3, q_4, q_5$ are the joint variables of joints 2,3,4 and 6, respectively, $d_5$ is the offset distance of joints 4 and 5, respectively, (as per the Denavit–Hartenberg notation) $L_2$ is the length between joints 1 and 2 and $L_3$ the length between joints 2 and 3, respectively, (as per the Denavit–Hartenberg notation) (Figure 3).

The singularity locus of the UR-3 manipulator can be found by solving $|J| = 0$. The manipulator lies in a singularity when:

$$\sin q_5 = 0 \tag{5}$$

$$q_3 = k\pi \text{ or } \sin q_3 = 0, \; k = 0, 1, 2, \dots \tag{6}$$

$$d_5 \cos(q_2 + 2q_3 + q_4) - L_2 \sin(q_2 + q_3) - d_5 - d_5 \cos(q_2 + q_4) + L_3 \sin q_2 + L_2 \sin(q_2 - q_3) - L_3 \sin(q_2 + 2q_3) = 0 \tag{7}$$

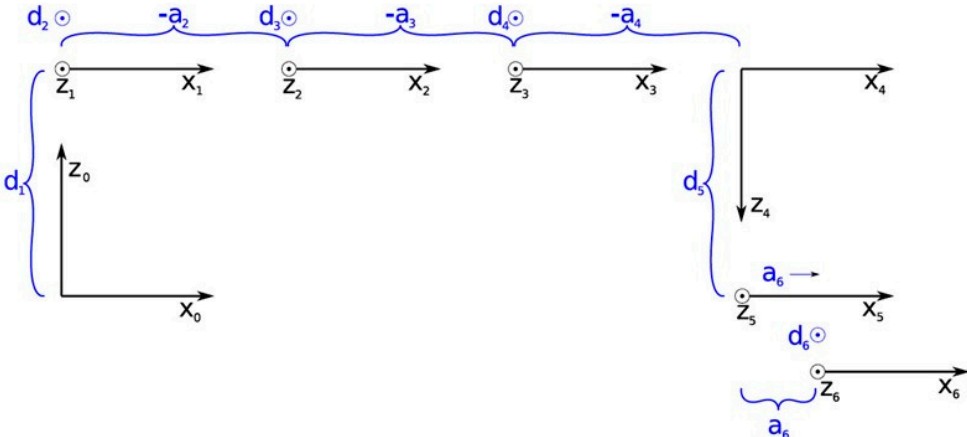

**Figure 3.** The Denavit–Hartenberg parameters notation for the UR3 manipulator.

The UR3 manipulator has eight IKP solutions included in each of the eight aspects of the configuration space defined by $[q_2\ q_3\ q_4\ q_5]$. In every aspect the $|J|$ sign is preserved. For two target frames that the manipulator must move from one to the other, a total of 64 paths are available, based on the combination of the possible postures at each point as depicted by the inverse kinematics solutions. However, only eight of these paths allow the motion of the robot from the first to the second frame to proceed while not meeting a singularity point in the joint space. The determination of the singular free paths is based on the signs shown in Table 1. The signs of all solutions of every frame are determined and allocated to the corresponded aspect.

**Table 1.** Rules for determining the aspects.

| Aspect No. | Sign of Equations (5)–(7) |
|:---:|:---:|
| 1 | 1, 1, 1 |
| 2 | 1, 1, −1 |
| 3 | 1, −1, 1 |
| 4 | 1, −1, −1 |
| 5 | −1, 1, 1 |
| 6 | −1, 1, −1 |
| 7 | −1, −1, 1 |
| 8 | −1, −1, −1 |

The singular free paths are those that the initial and final point in the configuration space have the same sign, or alternatively, are included to the same aspect.

### 2.1.2. Optimal Task Position Determination

The objective function is therefore calculated as the maximum minimum value of the MMA in the k points of the eight paths:

$$R_m\left(^B\mathbf{p}_L, \underset{-}{\theta}\right) = \begin{cases} \underset{j}{max}(\underset{i}{max}\left(r_{mi,j}\left(^B\mathbf{p}_L, \underset{-}{\theta}\right)\right)) \\ 0, \quad if\ \exists \left|J_{mi,j}\left(^B\mathbf{p}_L, \underset{-}{\theta}\right)\right| = 0 \end{cases} \tag{8}$$

where $i = 1, \dots, k, j = 1, \dots, 8$.

Moreover, the solution to the optimal task location search problem is given as:

$$\left(^B\mathbf{p}_L, \underset{-}{\theta}\right) = \mathrm{arg}max\left(R_m\left(^B\mathbf{p}_L, \underset{-}{\theta}\right)\right) \tag{9}$$

As such the optimization problem output is the position and orientation of {**L**} relative to {**B**}. Given that the desired end effector positions and orientations at the other task frames relative to {**L**} are a priori known, their derivation for the optimal task placement with respect to the robot's base is a simple transformation calculation.

### 2.2. Optimal Task Placement Determination

The layout design part of the proposed method undertakes the optimal task location determination for the given task. Initially, the path segments for the motion between frames are defined, given the sequence in which the end effector is to reach them. Intermediate frames are then defined for each path segment. The number of frames along the path is chosen by the designer, and depends on the task type and computational time and power limitations. Once the intermediate frames are determined, the search for the optimal task location is conducted employing a genetic algorithm (GA).

The application of the GA demands an appropriate form of representation of the optimization problem variables. In this case, a six digit chromosome of real numbers is selected, containing the position [x,y,z] coordinates of the origin point of {**L**} relative to {**B**} and the Z-Y-Z Euler angles depicting the orientation of {**L**} relative to {**B**} derived in Equation (1):

$$\underbrace{x \quad y \quad z}_{position} \quad \underbrace{\theta_z \quad \theta_y \quad \varphi_z}_{orientation} \tag{10}$$

The value ranges for the chromosome's genes were: $x \in [0.1, 0.5]$, $y \in [0.1, 0.5]$, $z \in [0.1, 0.5]$, in meters and $\theta_z = 0, \theta_y = 0, \varphi_z \in \left[-\frac{\pi}{2}, -\frac{\pi}{2}\right]$.

The following genetic operators were used.

Reproduction: The chromosomes of the current generation are reproduced in the next generation in an elitist manner, where the best individuals are guaranteed to survive to the next generation. In this work, 60% of the population is selected to survive to the next generation.

Crossover: The one-point crossover operator is selected for a part of the generation which is randomly selected using the crossover rate. Pairs of the individuals are swapping their parts defined by a randomly selected index in chromosomes. In this work, 20% of the population is selected for the crossover operator.

Mutation: A single point mutation is used for the rest individuals of the generation. A single point in the chromosome is randomly selected changed by adding a small random real number. In this work, 20% of the population is selected for the mutation operator.

The objective function was presented in Equation (6), and the GA's fitness function subject to minimization is as follows:

$$fitness = \frac{1}{R_m(^B\mathbf{p}_L, \underset{\sim}{\theta})} \tag{11}$$

The termination conditions that used are the maximum number of generations (400) and the maximum number (10) that a single generation is not improving (stalling generation). The population of every generation is (100).

Using the known reference of the task frames with respect to {**L**}, the position and orientation of each of the task frames relative to {**B**} are determined.

The output of the design part is the optimal location and orientation of all the task frames relative to {**B**}. These are also the input to the experimental implementation (Figure 1), where the experimental set-up's manipulator undertakes the considered task placed at the determined optimal location. The final output of the experimental implementation, of the proposed method is the manipulator's joint currents used during task execution.

### 3. Case Study Dynamic Task Optimal Location Derivation

A 2-D Lissajous curve was considered as the path that the manipulator's end effector had to follow in an analogous way as drawing the curve and exerting a force in the direction of the z-axis of {**T**}. The trajectory was performed by the human operator and is intermediate frames were defined by tracking the human operator's hand motion. The determined frames of the curve where then used to define the curve computationally for the purpose of the simulation process.

To follow the trajectory, a number of task frames were placed along the trajectory, where the first one was placed at the center of the curve, where {**L**} was considered to be placed, while the other local coordinate systems (frames) were referenced to {**L**}, presenting the same orientation and at distances determined using the straight-line segments starting points coordinates.

The considered trajectory was derived according to the proposed method. For comparison, a respective "bad" task location was also derived for both tasks, by simply interrupting the optimal search once the first non-singular task placement was derived. The path location for the optimal and "bad" locations are presented in Figure 4.

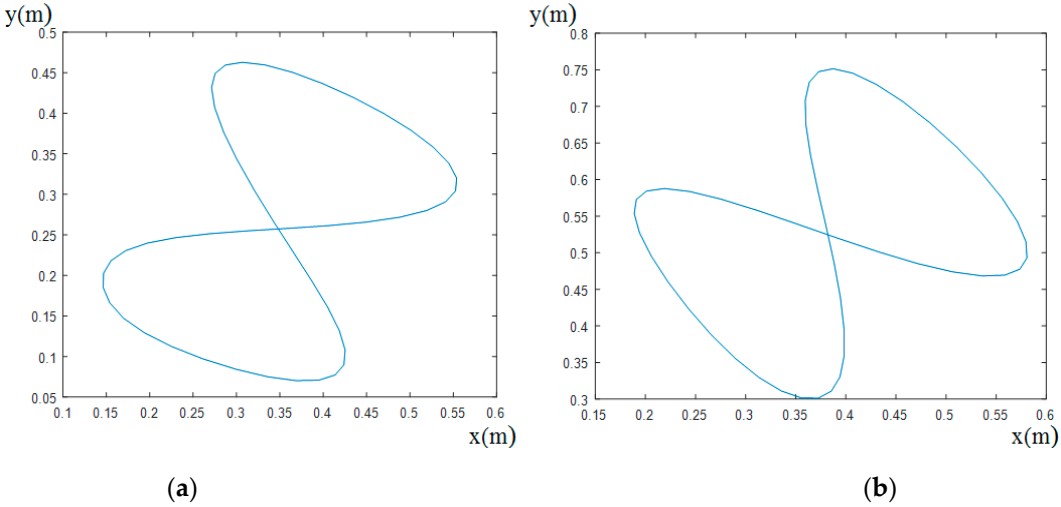

(**a**) (**b**)

**Figure 4.** Locations of (**a**) optimal and (**b**) "bad" task placement within the UR-3 manipulator's workspace.

The number of intermediate frames along the trajectory during the optimal placement determination was k = 50; however, in order to better approximate the curve an additional number of 20 frames was used along each path segment. The determined locations of the initial task frames {**L**} of the path for the optimal derived location and the "bad" location in joint space are presented in Table 2.

**Table 2.** Path starting frame {**L**} locations as derived in the joint space (in rad).

| Location | Optimal | "Bad" |
|----------|---------|-------|
| q1 | 3.726696 | 3.90893 |
| q2 | −1.276 | −0.68007 |
| q3 | 0.989997 | 0.78917 |
| q4 | −1.2848 | −1.6799 |
| q5 | −1.5708 | −1.5708 |
| q6 | −2.15269 | −1.55386 |

### 4. Experimental Verification of the Results

#### 4.1. Human Robot Interaction

Human Robot Interaction (HRI) was realized in the work to define and generate the trajectory for the robot's end-effector to perform the optimal placement of the robotic task in

the robot's working space. To this end the human operator teaches the robot by generating the trajectory to be followed that represents the task. The teaching of the manipulator utilized a process performed in Robotics Systems Lab., Bialystok University of Technology (BUT), Poland (Figure 5). To register human hand motion and generate the trajectory to follow by UR3 robot, the TrakSTAR electro-magnetic six-DoF tracker was used, which allows high position and orientation accuracy, i.e., 1.4 mm RMS (Root Mean Square) and 0.5 degrees RMS.

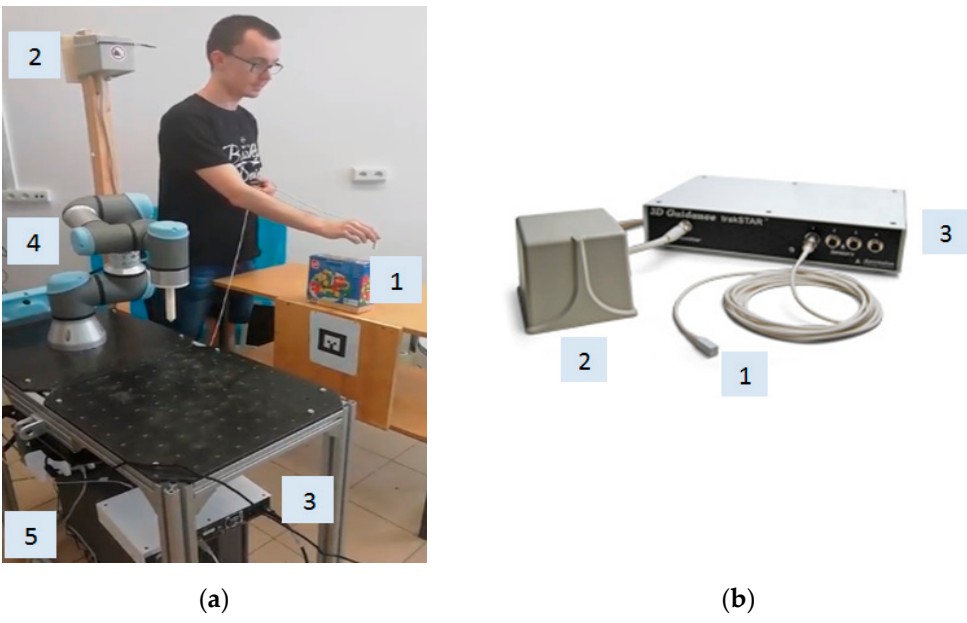

(**a**)  (**b**)

**Figure 5.** Human robot interactions: (**a**) human operator holds and moves TrakSTAR sensor (1) along the trajectory being generated for UR3 robot (4) to follow; (**b**) TrakSTAR electro-magnetic 6-DoF tracker, where (1) is the TrakSTAR sensor, (2) the Mid-Range Transmitter, (3) TrakSTAR Electronics Unit, (4) UR3 manipulator, (5) UR3 control box.

To perform the experiments with the UR3 robot in the laboratory environment, the magnetic Mid-Range Transmitter (MRT) of TrakSTAR was placed at the top platform of the mobile base close to the UR3 manipulator, while the TrakSTAR Electronics Unit connected by a cable with the TrakSTAR sensor was placed close to the UR3 control box mounted on the aluminum frame of the setup. For the generation of the trajectory for robot to follow, the human operator was holding the TrakSTAR sensor and moving it along the predefined trajectory in 3D or 2D space—in the last case the Z coordinate of the trajectory points was fixed in order for the motion to occur in the x–y plane.

The generated trajectory was registered by TrakSTAR at the frequency of 255 Hz and fed to UR3 robot, i.e., all trajectory points were saved in the robot's computer memory. To perform the trajectory following task with UR3, the trajectory was interpolated by cubic splines, the inverse kinematics task was performed at each point of the learned trajectory and the calculated manipulator joints angles were sent to the UR3 control unit and manipulator servomotors. The robot control was achieved through a built-in UR position control loop that runs with a frequency of 125 Hz. The trajectory generation, i.e., learning from demonstration task executed by the human operator, can be repeated again for the same or another trajectory type necessary for the optimal placement of the robotic task.

### 4.2. Optimal Task Location Determination

Human Robot Interaction (HRI) was realized in the work to define and generate the trajectory for robot's end-effector to perform the optimal placement of the robotic task in the robot's working space. The task of the optimal location experiment was performed at the Robotic Systems Laboratory, at Bialystok University of Technology (BUT), Poland.

The setup consists of a UR3 robot arm equipped with OnRobot gripper, HEX force torque sensor, UR3 controller, the control workstation with integrated program system and the PolyScope control panel. The UR3 manipulator is shown mounted on the setup in Figure 6.

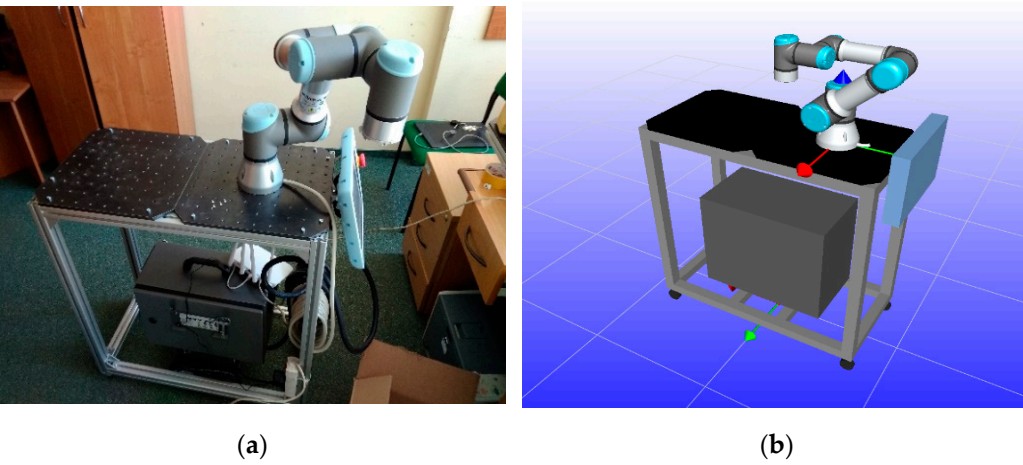

(**a**)                                    (**b**)

**Figure 6.** (**a**) UR3 robot; (**b**) RobWorkStudio environment used to visualize the robotic setup and control the robot trajectory. The coordinate system connected with the UR3 robot base is shown in the visualization (**b**).

Regarding the manipulator control, the stack is based on the robot operating system (ROS) and the User Interface is realized as a RobWork plugin [14].

The scheme of all the subsystem interactions of the laboratory setup is presented in Figure 7. Here, the plugin module belongs to application level of the integrated program system, the universal robot units belongs to the hardware control level, and the RobWork and ROS units form the system level.

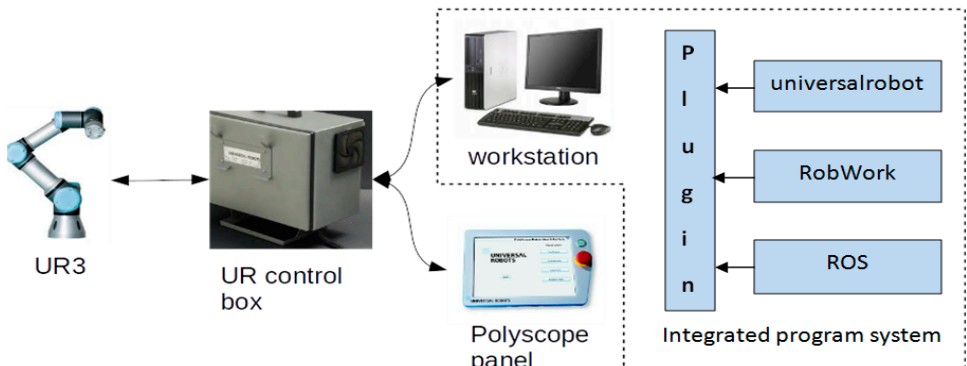

**Figure 7.** Subsystems interactions of laboratory setup.

For the trajectory following task, the movement of the robot was controlled by sending the joint velocity control signal to the control unit as an URScript command [15]. The control was updated with the rate of 125 Hz. The end effector velocity during task execution was constant. The URScript functions were accessed through the wrappers implemented in the universal robot packages. The control points of the trajectory were defined and visualized through the integrated program system based on the RobWork package. The subsystems were coordinated through the use of ROS (see Figure 6).

### 4.3. Trajectory Approximation

For the laboratory experiments, the UR3 manipulator end effector had to follow a trajectory inside its workspace, presenting the best possible dynamic performance (i.e., with minimum power consumption). Following the results of the computational part, the physical part of the experiment was executed in BUT Robotic Systems laboratory

environment. One of the aims of the physical task was to follow the predefined trajectory and measure the UR3 end effector and joint velocities as well as joint's angular change and the electrical current usage by each joint motor during the task execution for two cases, i.e., the task placed at the "optimal" and a "bad" locations.

The motor current was used to measure the robot's performance for the following reasons:

(a)  as shown in [5], joint torques are directly analogous to the motor current. As such the current magnitude may be utilized as a direct analogue to measuring the utilized joint torques.

(b)  Again, as shown in [5], measuring the utilized motor current also allows the monitoring of the manipulator's power consumption during task execution, since it was shown that it can be directly related to the joint torques and by extension of point (a) to the utilized current.

(c)  Although this process stems from a simplification of the manipulator's power usage, it has been shown in [5] that it provides a robust insight on the power utilization during task execution.

The end effector was programmed to accelerate and move with the constant tool velocity in the Cartesian space. In order for the end effector to not stop at each of the intermediate points along the path, a blend parameter termed radius with a value of 0.01 m was used.

### 4.4. Experimental Results

The resulting graphs of the electrical current utilized by the robot's joints as well as the total current utilized by the UR3 robot during task execution are presented in Figures 8–13.

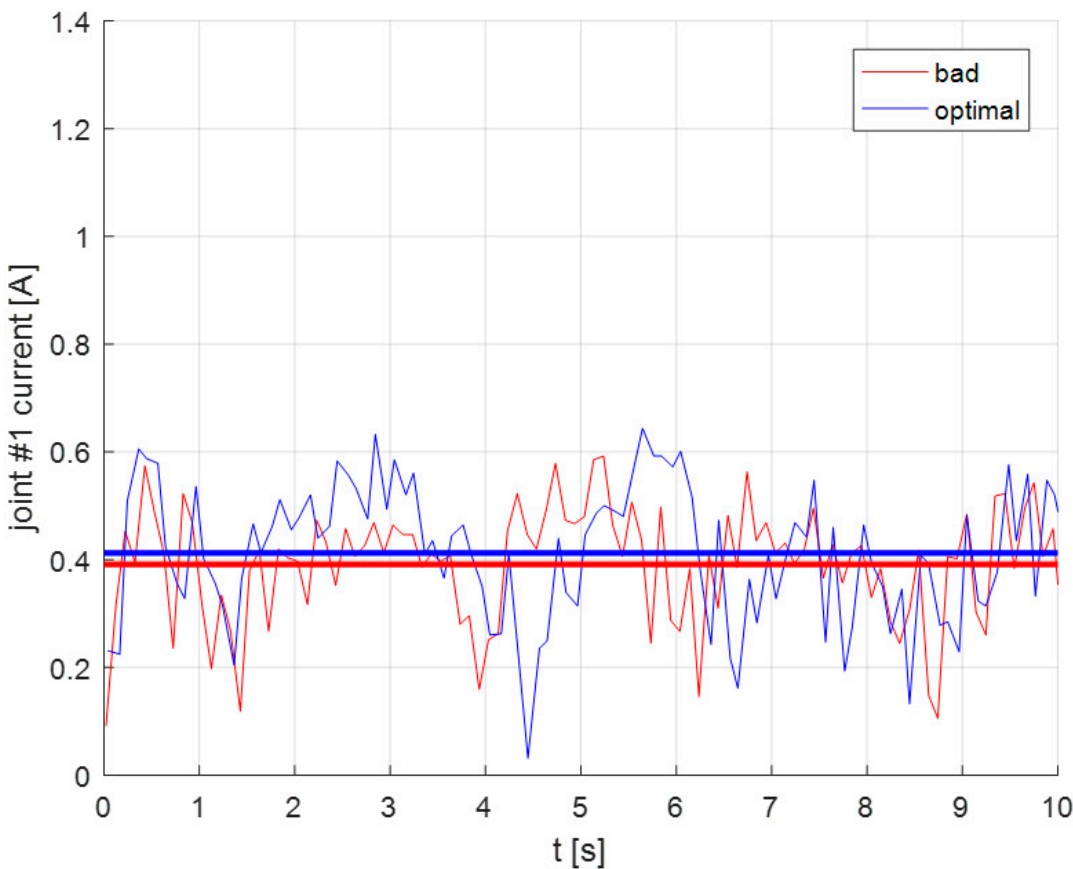

**Figure 8.** Current utilization and mean current during task execution in both locations for joint 1.

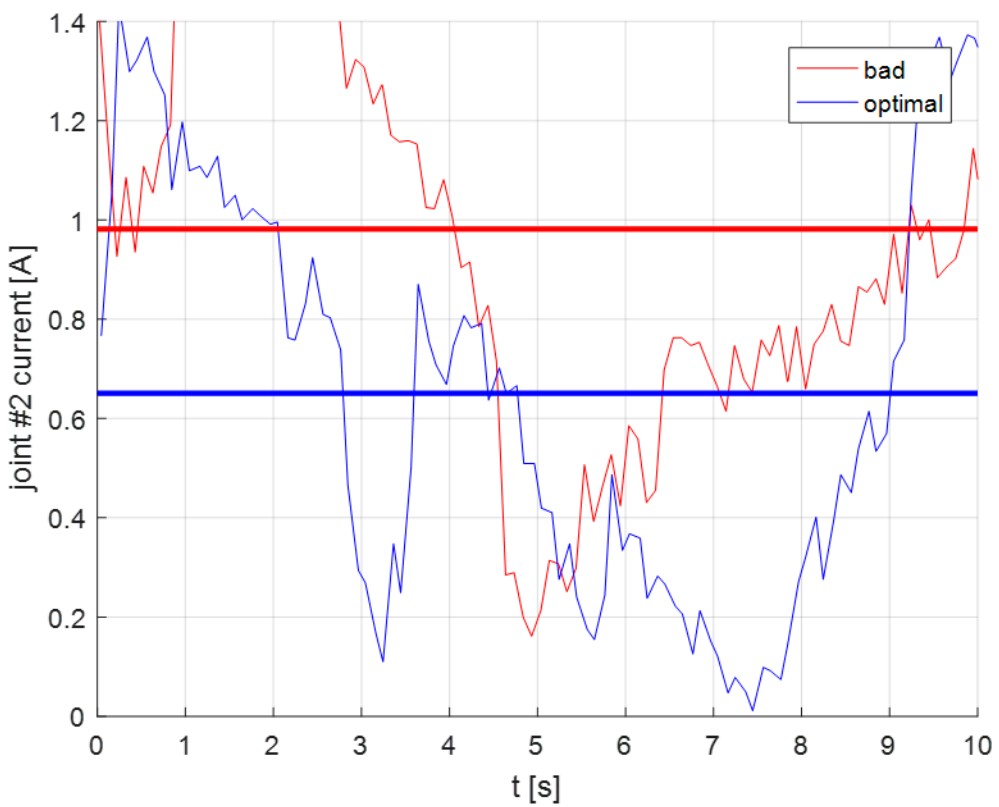

**Figure 9.** Current utilization and mean current during task execution in both locations for joint 2.

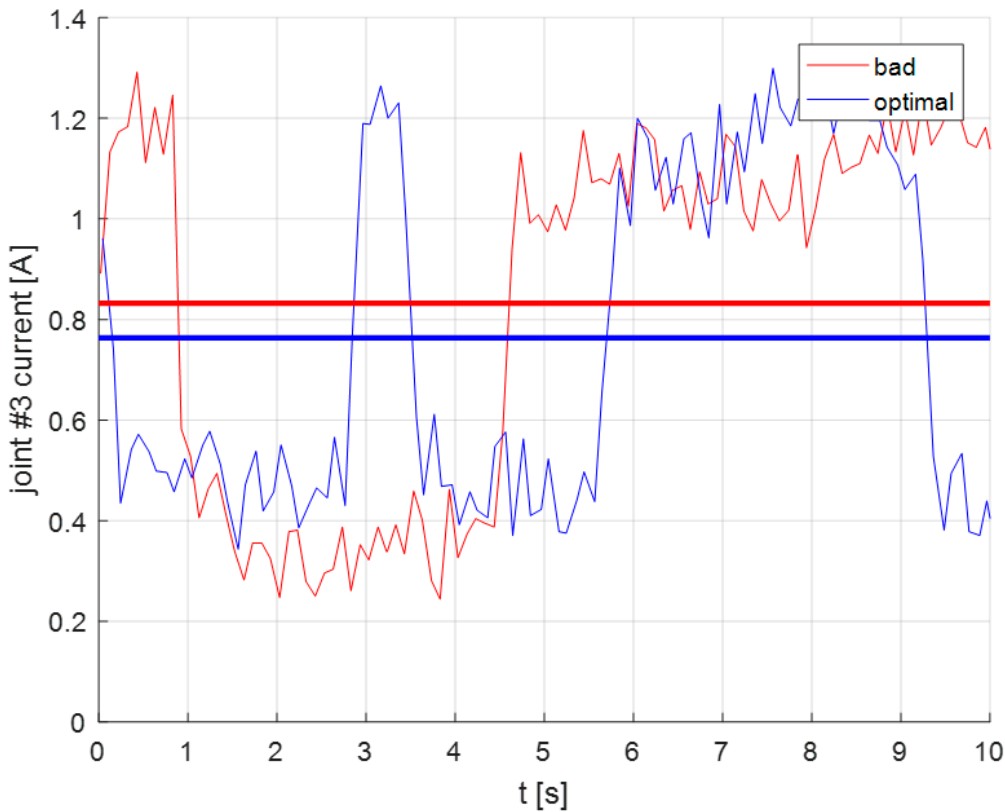

**Figure 10.** Current utilization and mean current during task execution in both locations for joint 3.

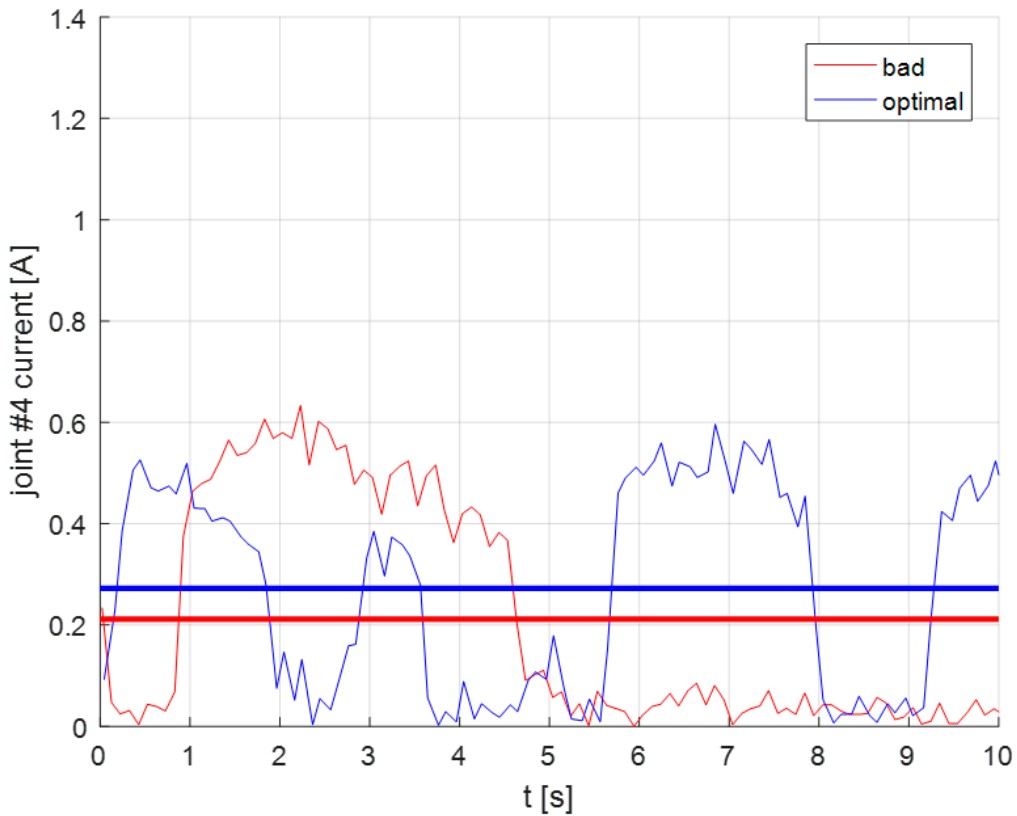

**Figure 11.** Current utilization and mean current during task execution in both locations for joint 4.

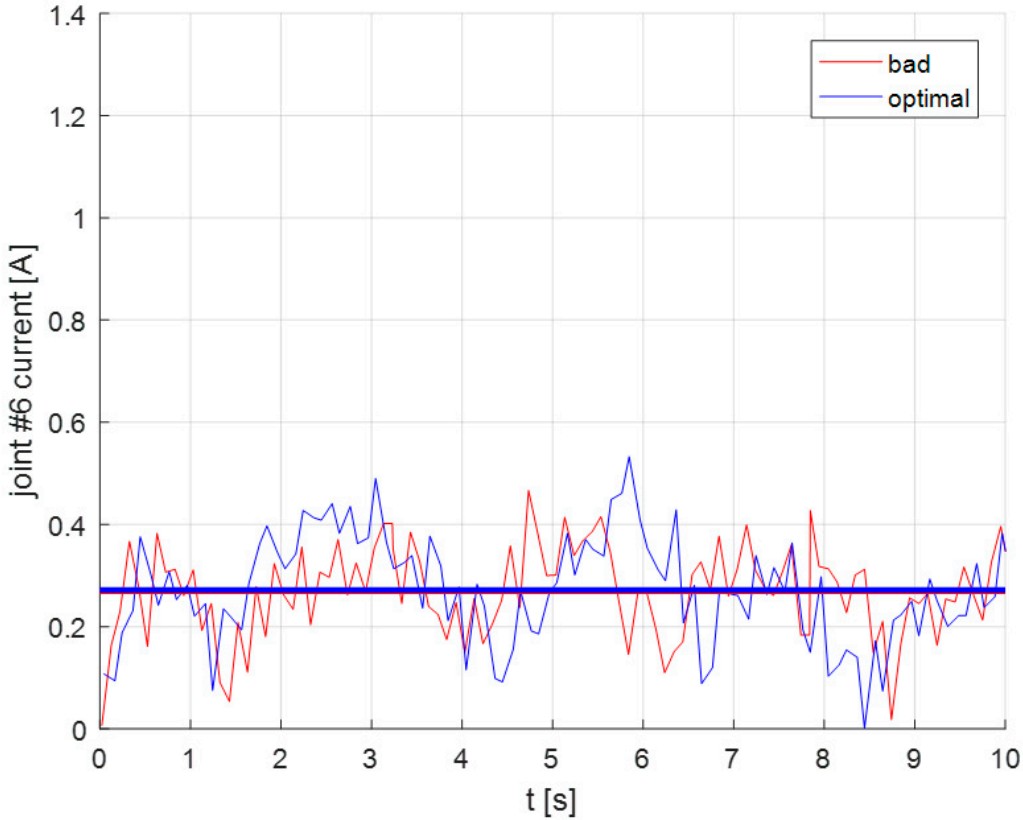

**Figure 12.** Current utilization and mean current during task execution in both locations for joint 6.

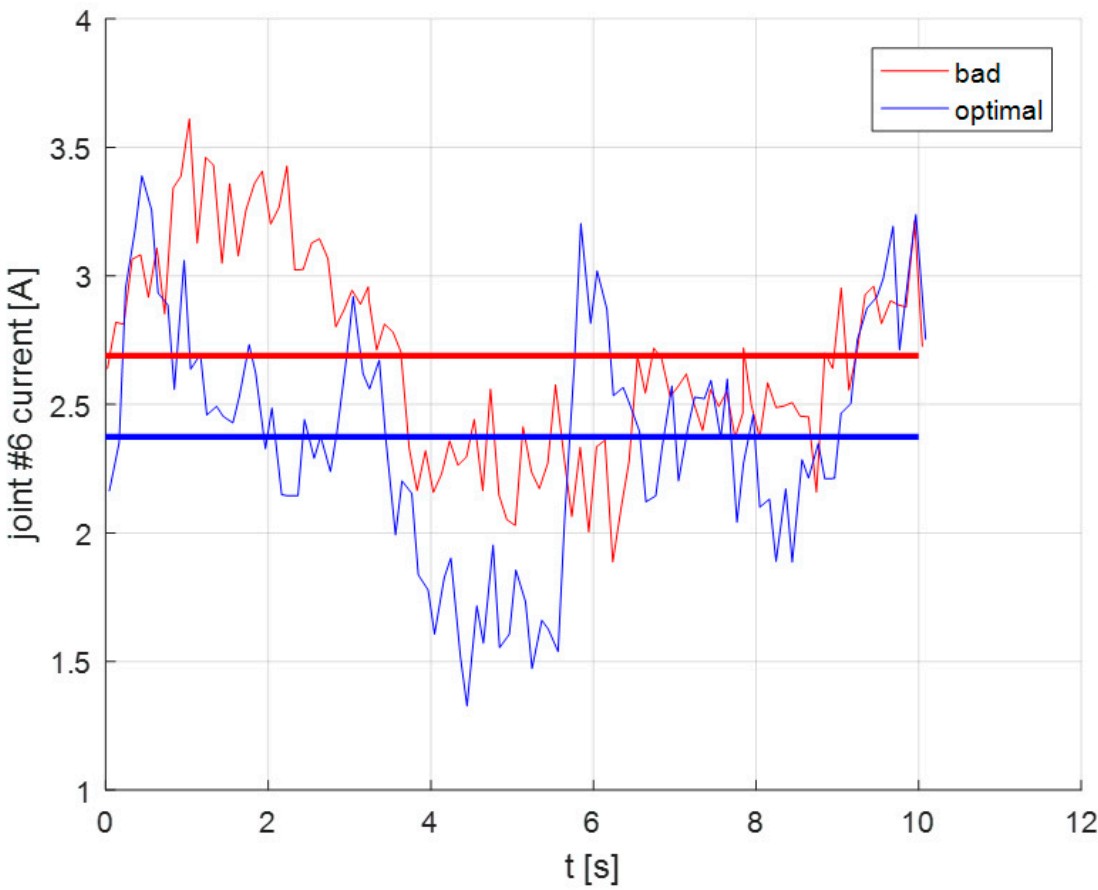

**Figure 13.** Total current utilization and total mean during task execution in both locations.

As depicted in the graphs presented in Figures 8–13, a reduction in the joint current utilization for the execution of the trajectory following the derived "optimal" task location under the same end effector velocities can be determined for two joints. The provided mean current line (whether for each joint or for the total current utilized by the robot during task execution) is also depicted and shows a significant decrease to the utilization of current by the manipulators joints 2 and 3, while the current utilization for joints 1 and 6 was almost identical and only joint 4 utilized higher mean current in the optimal location with respect to the "bad" location. Overall however the major point of interest is the limiting of the current utilized by joint 2 (as presented in Figure 9), which is the joint requiring significantly higher current values than the rest. The placement of the task in the optimal location has reduced the mean current utilized almost by 40%. The overall reduction in current utilization is depicted in the relative graph for the total current utilization (Figure 13) where a reduction in the range of about 12% can be determined for the derived "optimal" task placement. The total task execution time is about 10 s for both task placements. Joint 5 was not utilized by the robot for both locations and has been omitted from the results.

## 5. Discussion

As seen in Figures 7–12 the manipulator was able to perform the task using lower current at its joints and in total when the task was placed at the derived optimal solution as opposed to the "bad" one. The electrical current utilized is also a direct proportional indication of the utilization of the joint torques in order for the end effector to exert the specified force along the path and in the given direction. Given the MMA definition (as presented in paragraph 2) and the fact that both tasks were experimentally performed with the end effector exerting the same force along the path (in measure and direction), the main conclusions stemming from the results are the following:

- at the optimal task location, the manipulator requires less joint torques in order to exert the desired force along the path as opposed to the "bad" task location. Therefore, it is possible to deduce that the manipulator achieves a more efficient transmission of joint torques to the end effector applied force. This increased effectiveness in transmission may be interpreted into a number of different gains for the end user, such as lower energy consumption (which is also depicted in the graphs since the utilization of electrical current by the joints and the robot was measured), lower loading of joint's motors, etc.;
- moreover, since the process places the task in a location where the a given end effector force is exerted utilizing minimum joint torques, an increase in the joint torques at the optimal task location will result in the end effector exerting larger forces. This advantage allows the end user to place the task at a location that allows the full usage of its payload boundaries;
- the experimental results clearly verify the expected outcome of the optimal task placement method. This is a significant gain for the designer, since using the presented method they may place a dynamic task in a location where the robot will indeed exhibit increased dynamic performance via the minimization of utilized joint torques and increased energy consumption performance via the reduction in the utilized electrical current by the joint motors.

The optimal placement of the task allows the designer to choose on whether to perform the task with the minimum requirements in power consumption and joint loading or to significantly increase the forces exerted by the end effector. However, the advantage is that their choice could utilize the above advantages simultaneously, to some degree.

## 6. Conclusions

A method for the optimal task placement of robotic dynamic tasks in the manipulator's workspace was presented along with an experimental validation of the method's results using the UR3 robot. A dynamic task was considered as a case study application for the proposed methodology where the end effector had to exert a specific force (both in magnitude and in direction) along a 3D trajectory placed within the manipulator's workspace. A Human Robot Interaction (HRI) approach was chosen in the work to define the task, where the robot is taught the task trajectory by a human operator.

The dynamic index used to determine the optimal task location so as for the manipulator to achieve the highest possible dynamic performance during task execution was the manipulator mechanical advantage (MMA), which was used to formulate a task-based dynamic measure serving as the objective function of the optimization problem. The MMA provides the ratio of the end effector exerted force to the required joint toques.

The computational results from the presented method were verified by the respective experimental results that showed a significant reduction in joint torques for the execution of the considered task under the same end effector exertion of force. The optimal placement of the task allows the manipulator's end user to achieve a wide range of advantages and gains during task execution, such as the capability to execute a given dynamic task with minimum energy requirements and minimum joint motor load, while on the other end they may significantly reduce the task execution time.

Future work will include further experimental tests in order to determine the robot's true power consumption taking into account different aspects of its operation and the way these may inhibit the correct measurement of utilized current (as for example the use of a gear box in the joints, measurement sensitivity etc.) in order to provide greater accuracy.

Further future work will include an aggregation of kinematic and dynamic measures (such as the Manipulator Velocity Ration (MVR) [10] and the MMA) so as to determine the optimal location of a robotic task within the manipulator's work-space in order to achieve both the overall cycle time and joint motor strain reduction, along with the increase in the capacity of the robot to handle objects of large weights. The envisaged methodology will be used to optimally plan pick and place tasks of heavy objects within an industrial

environment. Moreover, future work will also involve applying the developed methods in a real-world scenario, where the task placement for tasks with different requirements may be planned and reconfigured online, for example through a cooperation with mobile robots.

**Author Contributions:** Conceptualization, N.A.; data curation, A.W.; formal analysis, K.M.; investigation, A.W., C.V., K.M. and V.M.; methodology, V.M.; software, C.V.; supervision, N.A.; visualization, A.W.; writing—original draft, C.V.; writing—review & editing, K.M. and V.M. All authors have read and agreed to the published version of the manuscript.

**Funding:** The work presented was partially financed by the Polish National Agency for Academic Exchange as part of the Academic International Partnerships (PPI/APM/2018/1/00033/U/001 project) and by the original research works WZ/WM-IIM/1/2019 and WZ/WE-IA/4/2020 funded by the Polish Ministry of Science and Higher Education. This research has been also financially supported by General Secretariat for Research and Technology (GSRT) and the Hellenic Foundation for Research and Innovation (HFRI) (Code: 1184).

**Conflicts of Interest:** The authors declare no conflict of interest.

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
