# Peer review of "Optimization of Dynamic Task Location within a Manipulator’s Workspace for the Utilization of the Minimum Required Joint Torques"

_electronics, doi:10.3390/electronics10030288_

Round 1

Reviewer 1 Report

In this paper, an optimization method is proposed for the manipulator operating path under the task space, which is expected to achieve accurate task planning with the minimum joint torque and/or consumed energy. But, from my point of view, this paper seems a incomplete experimental report with few discussions.

  1. The research status and existing problems were not fully understood in introduction.
  2. Author did not give reasonable derivation and proof. The experimental analysis was not compared with the existing international methods, and the specific and important steps were not clearly clarified. For example, what is the optimization process? What are the cutoff conditions for optimization? What is principle for selecting a particular point?
  3. What is the significance of the application of minimum joint torque, and what form is it presented in theoretically?
  4. What are the advantages and disadvantages compared with the low energy consumption or minimization cycle time optimization methods?
  5. In addition, in other words, even if the optimization gets a good effect, it does not elaborate the problem of work efficiency, which I think is also undesirable in the manufacturing industry.
  6. Finally, I think the diagram should be improved, especially Figure 4.

Author Response

Dear Reviewer,

Please, find our responses to your comments and suggestions in the attached PDF file. Thank you very much.

Best regards,

The Authors 

Reviewer 2 Report

The authors present an optimisation model of dynamic task location within a workspace of a manipulator. This study uses a Universal Robots UR-3 to validate the proposed model.

The proposed study has an important value for industrial application and they authors have used one of the commonest collaborative robot, UR-3, to validate their model.

Overall, the manuscript is well written and easy to follow. The analytical model is described as well as the experimental setup.

The review has the following minor suggestions to be addressed:

Point 1:

Figure 1 has a low quality. This should be replaced.

Point 2:

Line 188: Degree of freedom: should be degrees of freedom and the acronym inserted at this point of the manuscript, DOFs or DoFs.

Point 3:

Line 233 D.o.F. should be replaces with DoF or DOF. The “point” is not commonly used in robotics.

Information on the closed loop control is not reported, such as the sample time and frequency. It would be useful to insert this data, important for the stability of the control.

Point 4:

Conclusions section. All the experiments are reported by using information from the current of the joint and the total used by the robot. It would be nice to have a short description of limitations in this approach. Any error in the measurement to control the output force compared to the use of traditional force or torque sensors. For example, error in the measurement when a DC motor with a gear box is used, which is the case of the UR-3, sensitivity of the measurement, and overall advantages of this approach.

Point 5:

It would be nice to see a video of the robot in action.

Author Response

(The authors gave the same response as above.)

Reviewer 3 Report

The paper is an interesting one, but I have some recommendations for its improvement:

  1. All the used quantities have to be introduced in the paper text.
  2. The quantities used in Eq. (4) have to be highlighted on a schematic Figure.
  3. The quality of Figures 7-12 has to be improved.
  4. The results should be commented with more details.
  5. The References list has to be enlarged.

Author Response

(The authors gave the same response as above.)
